# Do LLMs Have ‘the Eye’ for MRI? Evaluating GPT-4o, Grok, and Gemini on Brain MRI Performance: First Evaluation of Grok in Medical Imaging and a Comparative Analysis

**DOI:** 10.3390/diagnostics15111320

**Published:** 2025-05-24

**Authors:** Alperen Sozer, Mustafa Caglar Sahin, Batuhan Sozer, Gokberk Erol, Ozan Yavuz Tufek, Kerem Nernekli, Zuhal Demirtas, Emrah Celtikci

**Affiliations:** 1Department of Neurosurgery, Sincan Training and Research Hospital, Ankara 06949, Turkey; a.sozer.md@gmail.com; 2Department of Neurosurgery, Kulu State Hospital, Konya 42780, Turkey; dr.mcaglarsahin@gmail.com; 3Ankara Medipol University Faculty of Medicine, Ankara 06050, Turkey; batuhansozer10@gmail.com; 4Department of Neurosurgery, Adiyaman Training and Research Hospital, Adiyaman 02100, Turkey; dr.gokberkerol@gmail.com; 5Department of Neurosurgery, Gazi University Faculty of Medicine, Ankara 06560, Turkey; ozantufek.md@gmail.com (O.Y.T.); zuhal.demrts@gmail.com (Z.D.); 6Department of Radiology, Stanford University School of Medicine, Stanford, CA 94305, USA; kerem.nernekli@gmail.com

**Keywords:** Gemini, GPT, Grok, large language model, magnetic resonance imaging, neuroradiology

## Abstract

**Background/Objectives**: Large language models (LLMs) are revolutionizing the world and the field of medicine while constantly improving themselves. With recent advancements in image interpretation, evaluating the reasoning capabilities of these models and benchmarking their performance on brain MRI tasks has become crucial, as they may be utilized—albeit off-label—for patient care by both neurosurgeons and non-neurosurgeons. **Methods**: ChatGPT-4o, Grok, and Gemini were presented with 35,711 slices of brain MRI, including various pathologies and normal MRIs. Models were asked to identify the MRI sequence and determine the presence of pathology. Their individual performances were measured and compared with one another. **Results**: GPT refused to answer 28.02% of the slices despite three attempts, whereas Grok and Gemini provided responses on the first attempt for every slice. Gemini achieved 74.54% pathology prediction and 46.38% sequence prediction accuracy. GPT-4o achieved 74.33% pathology prediction and 85.98% sequence prediction accuracy for questions that it had answered (53.50% and 61.67% in total, respectively). Grok achieved 65.64% pathology prediction and 66.23% sequence prediction accuracy. **Conclusions**: The image interpretation capabilities of the investigated LLMs are limited for now and require further refinement before competing with specifically trained and fine-tuned dedicated applications. Amongst them, Gemini outperforms the others in pathology prediction while Grok outperforms others in sequence prediction. These limitations should be kept in mind if use during patient care is planned.

## 1. Introduction

Artificial intelligence (AI) has made a large impact in every aspect of life in the past few years, including medicine and neurosurgical image interpretation. Although Cellular Neural Network systems were used in the early 2000s [1], this method has largely been abandoned in favor of Convolutional Neural Network (CNN) systems, which have been used for brain segmentation since 2015 [2]. In the field of neurosurgery, these radiological assessment capabilities of the AI are considered especially beneficial and showed promise for early warning systems in emergency settings [3]. Recently, AI applications in every field have gained great popularity when AI became a part of every household with large language models (LLMs).

Although LLMs usually use a very different and more recent method for image interpretation called Vision Transformer (ViT) [4], this new method is comparable to older and tested CNNs even now. With the integration of ViT into LLMs, image interpretation has rapidly become a central feature of major models in recent months.

ChatGPT-4o [5], xAI’s Grok [6], and Google’s Gemini (previously Bard) [7] all recently integrated image processing and interpretation capabilities. While their documentation states they are not validated to use for medical purposes, this does not preclude testing these capabilities in a research setting.

Especially in emergency departments and primary care settings, experienced neuroradiologists or neurosurgeons are not always readily available to interpret neuroradiological images. Moreover, even in urgent situations, image reporting can sometimes take hours and weeks in elective settings. This has driven the adoption of AI support in neurosurgical patient care across various settings, with numerous efforts dedicated to developing AI-based early warning systems. Widely accessible large language models (LLMs), with their intuitive and user-friendly interfaces, hold significant potential in this domain. Therefore, this study was conducted to assess the ability of these models to interpret brain magnetic resonance imaging (MRI) using a large-scale database, previously constructed from real-world data, for AI research.

## 2. Materials and Methods

### 2.1. Dataset Used

An expanded private version of previously published Gazi Brains 2020 Dataset [8] that contained segmented MRI data of real patients was used. An AI algorithm was developed using this dataset and was recently patented during the review process of this study [9]. This version contained fluid-attenuated inversion recovery (FLAIR), T2, T1 and T1c (contrast enhanced—when available) MRI scans of a total of 500 patients, including those who did not have any pathology (Normal, n = 93) and who had meningioma (n = 149), high-grade glioma (HGG, n = 60), cavernoma (n = 59), schwannomas (n = 38), post-operative patients who had various pathologies (n = 33), low-grade glioma (LGG, n = 30), metastatic disease (n = 19), and, classified as ‘Others’, a couple samples from various other pathologies (including: epidermoid tumors, hemorrhage, dermoid tumors, germinomas, pituitary tumors, arteriovenous malformations, hemangioblastomas, stroke patients, osteomas, plasmocytomas, and lymphomas, n = 19). A total of 1793 independent sequences and 35,711 slices were investigated. Slices that contained contrast-enhancing or non-enhancing parts of a tumor or lesion, peritumor edema, acute ischemia, encephalomalasic areas, cyst associated with a tumor, blood, or post-operative cavities were considered pathological and classified as ground truth positive. No ethical approval was necessary for this research as it was conducted using a pre-existing dataset according to local and regional guidelines.

### 2.2. Questioning LLMs

An OpenApi Software Development Kit (SDK) v.1.55.3 [10] was used to direct prompts to each LLM’s API. Image conversion to JPEG was performed using the OpenCV library with a quality setting of 95 in their original voxel resolution and converted to base64 format using the standard Python library (Python v.3.10.0) [11]. base64 formatted images were presented to LLMs with a standard prompt instructing the LLM as to what to do. The part of the code used that includes the prompt is presented in Appendix A. Other inference parameters were not explicitly set and were thus left to each model’s default behavior. All questioning was conducted between 20 November 2024 and 10 December 2024. None of the investigated services received any major update in this period to the best of the authors’ knowledge. The builds used were as follows: GPT-4o (gpt-4o-2024-08-06, released 6 August 2024), Gemini (gemini-1.5-pro-002, released 24 September 2024), and Grok (grok-vision-beta-0.1.0, released 23 November 2024). If an answer in the correct format was not provided, the algorithm was programmed to make 2 additional attempts before recording a “Not answered” status. Each slice was presented in an individual session with no memory function enabled to recall previous slices (i.e., every request was sent as a separate session), and the LLMs’ prediction regarding the presence of a pathology in the presented image and the prediction of the MRI sequence was recorded. Each LLM was tested in its native state without providing any additional training or fine-tuning.

### 2.3. Statistical Analysis

For the evaluation of the data obtained in this study and the statistical analysis, the “Statistical Package for Social Sciences” (SPSS) package program version 27.0 (Chicago, IL, USA) was used. Descriptive statistics were presented as numbers and percentages. When evaluating sequence predictions, even though both options were presented in the prompt, ‘T1’ and ‘T1c’ were considered correct reciprocally, since distinguishing them simply by assessing a single slice is not always possible. Results were aggregated at the ‘sequence-level’ to reflect clinical use. If any slice was predicted as pathological, the whole sequence was considered to be classified as pathological by that LLM, since there are pathologies actually visible in a couple of slices only. The differences between groups for categorical variables were evaluated using the Pearson Chi-square; results were reported and interpreted with adjusted standardized residuals (cut off: absolute 2.5). Fisher’s exact test was used for 2 × 2 tables. The Baptista–Pike method was used for odds ratio (OR) confidence intervals (95%CI); if the Baptista–Pike method could not be used, the Gart adjusted logit interval was used instead, and those results are marked with a ‘G’. Cochran’s Q test was used for comparing all three LLMs’ correct answer statuses together, and McNemar’s test with Bonferroni correction was used for individual (pairwise) comparisons between LLMs. Asymptotic significances are reported with χ^2^ statistics when the sample size adequacy assumption of this test was met, and exact *p* values are reported otherwise. Sensitivity, specificity, Positive Predictive Value (PPV), Negative Predictive Value (NPV), F Scores (F0.5: (precision weighted) − F1: (DICE) − F2: (recall weighted)) and accuracy (percentage correct) were calculated for each LLM [12]. Diagnostic values were evaluated using Receiver Operating Characteristic (ROC) analysis, and the area under the curves (AUCs) were calculated with confidence intervals. The results were evaluated with a confidence interval of 95.0% (95%CI) and a significance level of *p* ≤ 0.05; for post hoc analyses that require multiple comparisons, Bonferroni corrected *p* values are reported and evaluated.

## 3. Results

### 3.1. Dataset Descriptives and Answer Rates

Details of the analyzed dataset are presented in Table 1. Grok and Gemini provided a correctly formatted answer on the first attempt for every slice (number of requests sent = number of slices). However, GPT sometimes refused to provide a straightforward answer or sometimes completely refused to answer. GPT occasionally provided a valid response to a slice that it had previously refused to answer. Ultimately GPT refused to answer 10,006 slices (28.02%). A total of 71,542 requests were sent to GPT for the purposes of this study (2.003 requests/slice). Excluding 30,018 requests used for slices that were never answered by GPT, 41,524 requests were used to make GPT evaluate 22,581 slices (1.893 requests/slice), meaning GPT refused to answer 18,943 times for a slice that it eventually answered upon repeated requests. Reasons to refuse varied from ethical claims to confessing inability to decide (differing liberally between attempts, even for repeat attempts of the same slice; therefore, a systematic analysis of this refusal was not possible due to the probabilistic nature of this phenomenon). GPT was more likely to answer when the ground truth was positive [Fisher’s exact *p* < 0.001, OR = 1.19 (1.12–1.27)G]. GPT was considered to have answered incorrectly for the questions that it completely refused to answer when directly comparing with other LLMs. But individual metrics were calculated and presented both excluding those slices and including those slices as incorrect when possible.

### 3.2. Performance of GPT-4o

The performance of GPT was evaluated both by excluding the questions that it refused to answer and by considering them incorrect. A total of 33 (0.44%) T1c answers were considered correct for T1 sequence, and 3111 (41.42%) T1 answers were considered correct for T1c. Correct prediction of the sequence was associated with correct prediction of the pathology [Fisher’s exact *p* = 0.025], but the association was strangely inverse [OR: 0.91 (0.84–0.98)]. A confusion matrix that was based on the questions that it answered is presented in Table 2. Associations between correct pathology and sequence prediction were investigated separately for each sequence. Only the FLAIR sequence (lowest sequence prediction accuracy) showed an unexplained inverse association [Fisher’s exact *p* < 0.001, OR = 0.49 (0.43–0.55)]; T2 and T1 showed the expected positive correlation [Fisher’s exact *p* < 0.001 (both), OR = 1.48 (1.24–1.77), and OR = 1.51 (1.24–1.84), respectively], and the T1c sequence showed independence [Fisher’s exact *p* = 0.229]. Accuracy and sequence prediction accuracy were calculated as 53.50% (19,107/35,711) and 61.67% (22,023/35,711), respectively, when the slices for which it refused to answer were considered incorrect. GPT significantly performed differently for sequences (Adjusted residual) from best to worst as follows: T1 (7.7), FLAIR (6.6), T1c (−7.3), and T2 (−7.6) [χ^2^(3) = 159.746, *p* < 0.001].

### 3.3. Performance of Grok

The confusion matrix is presented in Table 3. Grok has clearly struggled to identify the FLAIR sequence (accuracy lower than random chance—25%), but its pathology prediction accuracy for this sequence was comparable to other sequences. For T1 sequence images, 135 (1.44%) T1c answers and 6022 (80.18%) T1 answers for T1c images were considered correct. In total, correct prediction of pathology seemed independent from sequence prediction [Fisher’s exact *p* = 0.064]; however, this was the result of the inverse association observed in FLAIR [Fisher’s exact *p* < 0.001, OR = 0.39 (0.35–0.43)G]; for T2 [Fisher’s exact *p* < 0.001, OR = 1.66 (1.50–1.83)], T1 [Fisher’s exact *p* < 0.001, OR = 1.73 (1.52–1.96)], and T1c [Fisher’s exact *p* < 0.001, OR = 1.39 (1.21–1.58)], correct prediction of the results was associated with correct identification of the sequence. Grok significantly performed differently for sequences (Adjusted residual) from best to worst as follows: T1 (5.7), T2 (1.5), FLAIR (−0.2), and T1c (−7.5) [χ^2^(3) = 70.356, *p* < 0.001].

### 3.4. Performance of Gemini

The confusion matrix is presented in Table 4. Gemini performed even worse than a die roll (25%) for predicting the FLAIR and T2 sequences but performed way better than random for T1 and T1c (their cut-off would be 50% because they are considered reciprocally correct). For T1 sequence images, 2420 (25.90%) T1c answers and 2820 (37.55%) T1 answers for T1c images were considered correct. Although in total, correct prediction of the pathology seemed independent from correct prediction of the sequence [Fisher’s exact *p* = 0.394], the FLAIR sequence showed an inverse association [Fisher’s exact *p* < 0.001, OR = 0.29 (0.26–0.32)], T2 showed independence [Fisher’s exact *p* = 0.972], and T1 and T1c showed positive correlation [Fisher’s exact *p* < 0.001 (both), OR = 3.51 (3.15–3.92), and OR = 2.63 (2.36–2.92), respectively]. Grok significantly performed differently for sequences (Adjusted residual) from best to worst as follows: T2 (7.3), FLAIR (5.6), T1 (−0.3), and T1c (−13.6) [χ^2^(3) = 208.026, *p* < 0.001].

### 3.5. Comparing LLMs and ROC Analyses

LLMs’ overall accuracy showed significant difference as the Cochran’s Q test shows [χ^2^(2) = 253,752.657, *p* < 0.001]. Post hoc analysis using McNemar’s test (with Bonferroni correction) showed that Grok is superior to GPT [χ^2^(1) = 1410.252, *p* < 0.001], and Gemini is superior to Grok [χ^2^(1) = 1085.637, *p* < 0.001] and GPT [χ^2^(1) = 4735.196, *p* < 0.001]. These outcomes were the same in every sequence individually (χ^2^ statistics omitted for readability).

Sequence prediction accuracy was significantly different between LLMs. Post hoc analysis showed that GPT is superior to Gemini even though slices for which GPT did not provide an answer were considered wrong; Grok is superior to Gemini and GPT. When sequences were analyzed individually, some deviations from this order was observed; therefore, the individual results and χ^2^ statistics are presented in Table 5.

GPT not answering to some slices created a challenge for ROC analysis. The first analysis was conducted excluding those slices for all LLMs (although, undoubtedly, these results would be biased in favor of GPT). AUCs (95%CI) were, respectively, 60.5% (59.8–61.2%), 57.9% (57.2–58.6%), and 57.8% (57.1–58.4%) for Gemini, Grok, and GPT, respectively. The AUC difference from Gemini was significant for Grok [z = 6.956, *p* < 0.001] and GPT [z = 7.948, *p* < 0.001]. The difference between Grok and GPT was not significant [z = 0.334, *p* = 0.738].

The second solution involved including those slices in some way. To do that, three different situations (answering positive for all of them [GPT (+)]; answering negative for all of them [GPT (−)], assigning them an intermediate unique value [GPT 0.5]) for GPT was evaluated. ROC Curves and Precision–Recall curves are presented in Figure 1. Individual AUCs and comparisons are presented in Table 6. Gemini significantly outperformed all, and GPT (+) was significantly outperformed by all. Assuming the intermediate value (0.5) resulted in the highest AUC for GPT.

### 3.6. Accuracy in Different Pathologies

A total of 6997 slices that contain at least one pathology were present in the dataset. GPT answered 5232 of them, and detailed accuracy distributions for different pathologies are presented in Table 7. LLMs’ accuracy showed significant difference [χ^2^(2) = 1009.917, *p* < 0.001]. Pairwise analysis showed Gemini is superior to GPT [χ^2^(1) = 329.376, *p* < 0.001]; Grok is superior to Gemini [χ^2^(1) = 223.779, *p* < 0.001] and GPT [χ^2^(1) = 885.939, *p* < 0.001]. These outcomes were the same in every pathology individually, except schwannoma (GPT vs. Gemini ns. after Bonferroni correction), cavernoma (GPT vs. Gemini ns.), and others (Grok vs. Gemini after Bonferroni correction) (χ^2^ statistics omitted for readability).

### 3.7. Slice Difficulty

None of the LLMs were able to correctly answer for 4975 (13.4%) slices, 11,482 (32.2%) were classified correctly by two LLMs at the same time, and 13.474 (37.7%) were correctly classified by all LLMs. Detailed distributions are presented in Figure 2. Representative MRI slices illustrating model predictions have been included as Appendix B to improve transparency.

### 3.8. Sequence-Level Analysis

Individual accuracies of all LLMs for various pathologies are presented in Table 8. There were no sequences that GPT completely refused to answer for any slices of, and its responses are based on the slices that it had answered.

Grok only classified one sequence of one patient as all-slices normal among 299 completely normal brain MRIs, showing extreme overestimation for this matter. This constant overestimation has falsely presented high accuracy for sequences that have actual pathology. Statistical tests were run separately for ground truth positives and negatives.

Ground truth positive sequence (n = 1480) performance was significantly different among LLMs [χ^2^(2) = 431.807, *p* < 0.001]. As post hoc analysis showed, Gemini is superior to GPT [χ^2^(1) = 100.363, *p* < 0.001], and Grok is superior to GPT [χ^2^(1) = 362.024, *p* < 0.001] and Gemini [χ^2^(1) = 152.910, *p* < 0.001].

Ground truth negative sequence (n = 313) performance was also significantly different among all LLMs [χ^2^(2) = 431.807, *p* < 0.001]. Additionally, Gemini is superior to Grok [χ^2^(1) = 100.363, *p* < 0.001], and GPT is superior to Gemini [χ^2^(1) = 7062, *p* < 0.024] and Grok [χ^2^(1) = 92.092, *p* < 0.001], as post hoc analyses indicate. Obviously, these results are biased in favor of GPT since it did not respond to some slices, reducing the possibility of misclassifying as positive.

GPT overestimated (i.e., classified more slices than the number actually pathological) in 418 (23.3%) sequences and underestimated (i.e., classified less) in 1097 (61.2%). Gemini overestimated in 651 (36.3%) and underestimated in 813 (45.3%), whereas Grok overestimated in 1331 (74.2%) and underestimated in 309 (17.2%).

## 4. Discussion

The results of this study indicate that Grok is the most accurate sequence predictor, while Gemini is the most reliable for pathology detection amongst the investigated LLMs for brain MRI interpretation. Although Grok clearly struggled to identify the FLAIR sequence, overall it performed well in identifying, especially for T2 and T1 sequences. Gemini was unreliable in identifying sequences, performing worse than random prediction for FLAIR and T2, but it was the most accurate and had the best AUC in the ROC analysis. Identifying FLAIR was an important strength of GPT when compared to others. Although, distinguishing FLAIR from T2 sequences may depend heavily on the anatomical region being imaged, particularly concerning the visibility of cerebrospinal fluid, which poses a potential limitation for both human readers and automated models. GPT’s unanswered slices were mostly ground truth negative slices, possibly due to security systems designed to avoid missing potentially important pathologies. Nevertheless, GPT is still outperformed by Gemini significantly (and Grok, although this is statistically insignificant) when those responses were considered intermediate or negative. Both of the other LLMs would be outperforming GPT, should those responses be positive. Grok has the highest accuracy in slices that contain actual pathology, showing higher sensitivity at the expense of reduced specificity.

A key challenge encountered in this study was the slices for which GPT did not provide an answer. This behavior was unique to GPT and required careful consideration. Considering all unanswered slices as incorrect would unfairly penalize GPT. The exact mechanism behind GPT’s refusal likely involves the rule-based reward system described in OpenAI’s official documentation [13]. Other investigated LLMs mention various safety systems but do not detail them (they also complied with every request sent for the purposes of the present study). However, even when unanswered slices were excluded from analysis (introducing severe bias favoring GPT), Gemini still demonstrated better performance in the ROC analysis. It should be noted that although differences between models were statistically significant, all AUC values remained relatively low (below 60%), indicating limited overall discriminative power. The sharp drop in Precision–Recall curves at higher recall levels likely reflects class imbalance, given that normal slices outnumber abnormal ones by approximately four to one. Another limitation was the binary classification system. Since not all pathologies are clearly visible on every MRI sequence, the only reliable question for single-slice analysis was whether the slice was normal or abnormal. We addressed this limitation by grouping slices according to the specific pathology present and assessing them separately. No clinical information about patients was provided to any of the LLMs, as this study aimed to evaluate MRI interpretation capabilities rather than comprehensive clinical decision-making skills. This may have reduced the performance for all models. The importance of presenting clinical context may be assessed in dedicated studies.

Grok’s real time integration with its social media platform X [6] makes it unique, adding extra interest to its results and comparisons to other LLMs. However, reports regarding the performance of Grok in the field of medicine are very limited, with only 3 relevant results among 14 total in PubMed. Two of those reports investigate patient information related capabilities, and one is about medical exam questions. None of these studies are actually investigating medical images or any image interpretation capabilities, placing them outside the scope for the purposes of the present study. This study is a first in this aspect for Grok.

There are no reports investigating Gemini alone. All existing reports compare Gemini with GPT and other LLMs. Among those, only Günay et al. provided actual clinical images (of ECGs) and asked the LLM to interpret the image [14]. Although they concluded that the LLMs’ performance is not adequate to rely on them, their results also show that GPT-4o is superior to Gemini Advanced. Another real image interpretation study was the Hindy et al. study that investigated gram-stained bacteria identification capabilities, concluding that GPT-4o outperforms Gemini 1.5 Pro; although, both LLMs are unreliable [15]. In the field of neuroradiology, Gupta et al. compared GPT-4 to its predecessor GPT-3.5 and Gemini for text based multiple choice questions, and although GPT-4 outperformed the other two, its best accuracy remained at 64.89% [16]. The other few studies neither contained actual medical images nor were related to neuroradiology, placing them outside the scope.

The most investigated LLM is perhaps GPT, with dozens of articles for each version of it. However, since the image interpretation is a recent feature, evidence regarding this capability is still very limited. One of the most relevant studies was by Zhang et al. [17]. They pre-trained GPT-4 with images of cerebral hemorrhages and then asked it to segment the bleeding areas. Even though they had pre-trained the model and presented the model only with the slices they deemed most representative of the condition, they concluded that GPT showed promising results, but it is not adequate just yet for use in a clinical setting. Pre-training appears to significantly improve the performance of GPT and other LLMs [18].

Two studies with some level of similarity in design investigated multimodal radiological diagnostic capabilities of GPT-4v (earlier version with image interpretation) both concluding that GPT excels at identifying the modality. Brin et al.’s study only included 43 head CT images (out of 103 total CT slices), and although head CT results were not presented separately, the overall diagnostic accuracy for CT was reported as 36.4% [19]. On the other hand, Strotzer et al. reported individual results, and GPT achieved 60% accuracy for MS, 56% for brain tumors, 54% for brain hemorrhage, and 46% for ischemic stroke in their multimodal study [20].

On the other hand, the current gold standard method for pathology prediction in MRI is specifically trained and fine-tuned dedicated models based on CNN or ViT. Recently, ViT-based models have outperformed CNN-based models with accuracies reported above 98–99% [21,22]. This recent report of ViT-based models indicates that the LLMs that use ViT as we discussed are easily able to adapt to these improvements to achieve similar results in the near future. Subsequently, LLMs may prove very useful and easy to use as triage support or decision support systems, once their reliability is established.

## 5. Conclusions

This is the first study to investigate the neuroradiology performance of ChatGPT-4o, Gemini and Grok each individually and in direct comparison. The image interpretation capabilities of these LLMs are limited for now and require improvement before they can compete with specifically trained and fine-tuned CNN- or ViT-based applications. Gemini outperformed others in pathology prediction (with the highest AUC), while Grok did better in sequence prediction tasks. Notably, Grok showed significantly higher accuracy than the other models when an actual pathology is present, though at the cost of reduced specificity. Future studies could leverage multi-class classification to provide a more clinically meaningful evaluation beyond the binary classification approach used here.

## Figures and Tables

**Figure 1 diagnostics-15-01320-f001:**
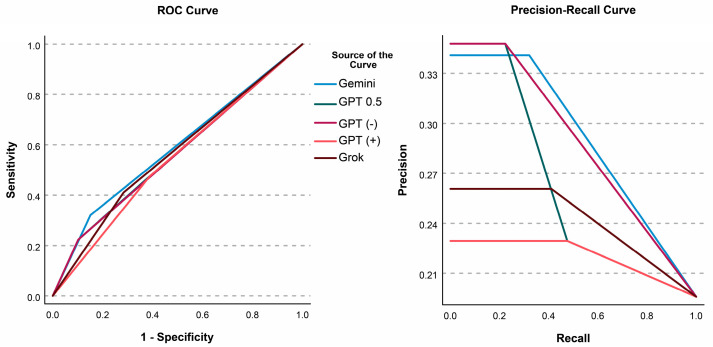
ROC Curve and Precision–Recall curve for all model variations. GPT 0.5 = missing data were replaced with an intermediate value (0.5); GPT (+) = missing answers were assumed as positive; GPT (−) = missing answers were assumed as negative.

**Figure 2 diagnostics-15-01320-f002:**
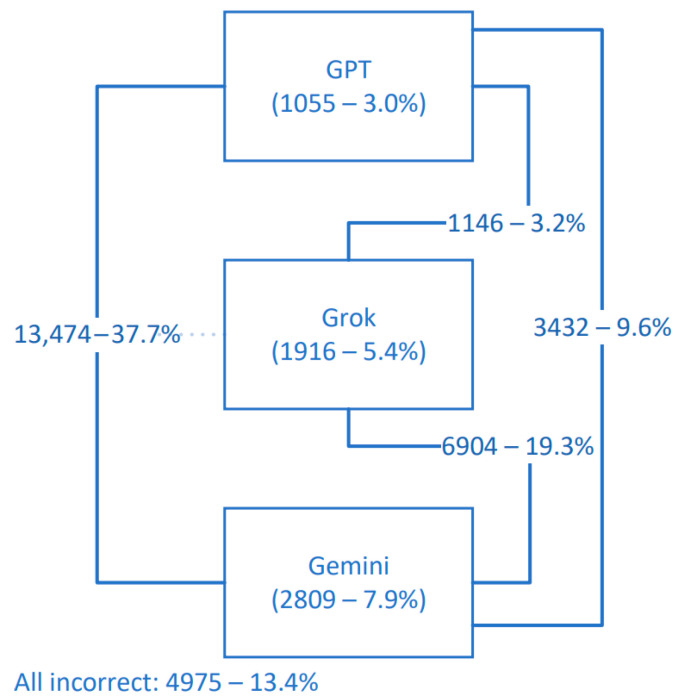
Correct classification rates of MRI slices by ChatGPT-4o, Grok, and Gemini, including model combinations and cases where all models failed.

**Table 1 diagnostics-15-01320-t001:** Descriptive information regarding the investigated dataset.

Origin Patient	Sequence									
	FLAIR (n = 9511, 26.63%)		T2 (n = 9347, 26.17%)		T1 (n = 9343, 26.16%)		T1c (n = 7510, 21.03%)		Total (n = 35,711)	
	Positive	Negative	Positive	Negative	Positive	Negative	Positive	Negative	Positive	Negative
Normal	0	1854 (100%)	0	1819 (100%)	0	1819 (100%)	0	429 (100%)	0	5921 (100%)
HGG	499 (40.54%)	732 (59.46%)	499 (40.94%)	720 (59.06%)	499 (41.%)	718 (59.%)	499 (41.07%)	716 (58.93%)	1996 (40.88%)	2886 (59.12%)
LGG	116 (20.21%)	458 (79.79%)	116 (20.42%)	452 (79.58%)	116 (20.42%)	452 (79.58%)	115 (20.32%)	451 (79.68%)	463 (20.34%)	1813 (79.66%)
Meningioma	577 (20.88%)	2187 (79.12%)	577 (21.27%)	2136 (78.73%)	577 (21.27%)	2136 (78.73%)	572 (21.72%)	2061 (78.28%)	2303 (21.28%)	8520 (78.72%)
Schwannoma	102 (14.55%)	599 (85.45%)	102 (15.04%)	576 (84.96%)	102 (14.96%)	580 (85.04%)	102 (14.96%)	580 (85.04%)	408 (14.87%)	2335 (85.13%)
Cavernoma	128 (12.46%)	899 (87.54%)	128 (12.62%)	886 (87.38%)	128 (12.67%)	882 (87.33%)	97 (12.29%)	692 (87.71%)	481 (12.53%)	3359 (87.47%)
Metastatic Disease	130 (34.39%)	248 (65.61%)	130 (34.57%)	246 (65.43%)	130 (34.76%)	244 (65.24%)	130 (34.67%)	245 (65.33%)	520 (34.6%)	983 (65.4%)
Other	84 (24.78%)	255 (75.22%)	84 (25.07%)	251 (74.93%)	84 (25.07%)	251 (74.93%)	71 (24.15%)	223 (75.85%)	323 (24.79%)	980 (75.21%)
Post-op	130 (20.22%)	513 (79.78%)	129 (20.64%)	496 (79.36%)	129 (20.64%)	496 (79.36%)	115 (21.82%)	412 (78.18%)	503 (20.79%)	1917 (79.21%)
TOTAL	1766 (18.57%)	7745 (81.43%)	1765 (18.88%)	7582 (81.12%)	1765 (18.89%)	7578 (81.11%)	1701 (22.65%)	5809 (77.35%)	6997 (19.59%)	28,714 (80.41%)

**Table 2 diagnostics-15-01320-t002:** Performance metrics and confusion table of GPT-4o. Care should be taken interpreting this table, since this excludes slices that the GPT refused to answer. TP: True Positive, TN: True Negative, FP: False Positive, FN: False Negative, PPV: Positive Predictive Value, NPV: Negative Predictive Value, LR: Likelihood Ratio (Positive or Negative), DOR: Diagnostic Odds Ratio.

	TP	TN	Sensitivity	Specificity	LR+	DOR	F1 (DICE)	Accuracy	Sequence Accuracy
	FP	FN	PPV	NPV	LR−	F0.5	F2		
FLAIR	643 (8.32%)	4720 (61.11%)	0.428	0.759	1.776	2.355	0.353	69.43%	5652/7724 (73.17%)
	1501 (19.43%)	860 (11.13%)	0.300	0.846	0.754	0.319	0.394		
T2	354 (5.80%)	4334 (70.98%)	0.291	0.887	2.575	3.223	0.333	76.78%	5469/6106 (89.57%)
	554 (9.07%)	864 (14.15%)	0.390	0.834	0.799	0.365	0.307		
T1	129 (1.92%)	5189 (77.14%)	0.100	0.955	2.222	2.359	0.155	79.05%	6727/7285 (92.34%) ^1^
	242 (3.60%)	1167 (17.35%)	0.348	0.816	0.942	0.233	0.117		
T1c	435 (8.45%)	3303 (64.16%)	0.358	0.840	2.238	2.929	0.382	72.61%	4733/5148 (91.94%) ^1^
	629 (12.22%)	781 (15.17%)	0.409	0.809	0.764	0.398	0.367		
TOTAL	1561 (6.07%)	17,546 (68.26%)	0.298	0.857	2.084	2.545	0.321	74.33%	22,581/26,263 (85.98%)
	2926 (11.38%)	3672 (14.29%)	0.348	0.827	0.819	0.337	0.307		

^1^ T1 and T1c answers were considered interchangeably.

**Table 3 diagnostics-15-01320-t003:** Performance metrics and confusion table of Grok. TP: True Positive, TN: True Negative, FP: False Positive, FN: False Negative, PPV: Positive Predictive Value, NPV: Negative Predictive Value, LR: Likelihood Ratio (Positive or Negative), DOR: Diagnostic Odds Ratio.

	TP	TN	Sensitivity	Specificity	LR+	DOR	F1 (DICE)	Accuracy	Sequence Accuracy
	FP	FN	PPV	NPV	LR−	F0.5	F2		
FLAIR	847 (8.91%)	5388 (56.65%)	0.480	0.696	1.579	2.114	0.341	65.56%	1610/9511 (16.93%)
	2357 (24.78%)	919 (9.66%)	0.264	0.854	0.747	0.290	0.413		
T2	762 (8.15%)	5432 (58.11%)	0.432	0.716	1.521	1.918	0.326	66.27%	7314/9347 (78.25%)
	2150 (23.00%)	1003 (10.73%)	0.262	0.844	0.793	0.284	0.382		
T1	505 (5.41%)	5852 (62.64%)	0.286	0.772	1.254	1.356	0.252	68.04%	8231/9343 (88.10%) ^1^
	1726 (18.47%)	1260 (13.49%)	0.226	0.823	0.925	0.236	0.272		
T1c	759 (10.11%)	3895 (51.86%)	0.446	0.671	1.356	1.642	0.347	61.97%	6495/7510 (86.48%) ^1^
	1914 (25.49%)	942 (12.54%)	0.284	0.805	0.826	0.306	0.400		
TOTAL	2873 (8.05%)	20,567 (57.59%)	0.411	0.716	1.447	1.758	0.319	65.64%	23,650/35,711 (66.23%)
	8147 (22.81%)	4124 (11.55%)	0.261	0.833	0.823	0.282	0.369		

^1^ T1 and T1c answers were considered interchangeably correct.

**Table 4 diagnostics-15-01320-t004:** Performance metrics and confusion table of Gemini. TP: True Positive, TN: True Negative, FP: False Positive, FN: False Negative, PPV: Positive Predictive Value, NPV: Negative Predictive Value, LR: Likelihood Ratio (Positive or Negative), DOR: Diagnostic Odds Ratio.

	TP	TN	Sensitivity	Specificity	LR+	DOR	F1 (DICE)	Accuracy	Sequence Accuracy
	FP	FN	PPV	NPV	LR−	F0.5	F2		
FLAIR	606 (6.37%)	6688 (70.32%)	0.343	0.864	2.522	3.318	0.353	76.69%	2135/9511 (22.45%)
	1057 (11.11%)	1160 (12.20%)	0.364	0.852	0.760	0.360	0.347		
T2	572 (6.12%)	6658 (71.23%)	0.324	0.878	2.656	3.449	0.351	77.35%	1340/9347 (14.34%)
	924 (9.89%)	1193 (12.76%)	0.382	0.848	0.770	0.369	0.334		
T1	396 (4.24%)	6557 (70.18%)	0.224	0.865	1.659	1.849	0.248	74.42%	7618/9343 (81.54%) ^1^
	1021 (10.93%)	1369 (14.65%)	0.279	0.827	0.897	0.266	0.233		
T1c	669 (8.91%)	4473 (59.56%)	0.393	0.770	1.709	2.169	0.361	68.47%	5468/7510 (72.81%) ^1^
	1336 (17.79%)	1032 (13.74%)	0.334	0.813	0.788	0.344	0.380		
TOTAL	2243 (6.28%)	24,376 (68.26%)	0.321	0.849	2.126	2.658	0.331	74.54%	16,561/35,711 (46.38%)
	4338 (12.15%)	4754 (13.31%)	0.341	0.837	0.800	0.337	0.325		

^1^ T1 and T1c answers were considered interchangeably correct.

**Table 5 diagnostics-15-01320-t005:** Sequence prediction accuracy comparison results.

	Cochran’s Q	GPT vs. Grok	GPT vs. Gemini	Grok vs. Gemini	LLM Order
FLAIR	χ^2^(2) = 4863.681 *p* < 0.001	χ^2^(1) = 3365.557, *p* < 0.001	χ^2^(1) = 2566.381, *p* < 0.001	χ^2^(1) = 122.088, *p* < 0.001	GPT > Gemini > Grok
T2	χ^2^(2) = 7041.433 *p* < 0.001	χ^2^(1) = 1031.342, *p* < 0.001	χ^2^(1) = 2924.384, *p* < 0.001	χ^2^(1) = 5229.658, *p* < 0.001	Grok > GPT > Gemini
T1	χ^2^(2) = 1588.105 *p* < 0.001	χ^2^(1) = 1358.836, *p* < 0.001	χ^2^(1) = 628.697, *p* < 0.001	χ^2^(1) = 186.248, *p* < 0.001	Grok > Gemini > GPT
T1c	χ^2^(2) = 1156.113 *p* < 0.001	χ^2^(1) = 1146.864, *p* < 0.001	χ^2^(1) = 171.306, *p* < 0.001	χ^2^(1) = 461.498, *p* < 0.001	Grok > Gemini > GPT
TOTAL	χ^2^(2) = 196,901.084 *p* < 0.001	χ^2^(1) = 189.132, *p* < 0.001	χ^2^(1) = 1741.563, *p* < 0.001	χ^2^(1) = 3759.616, *p* < 0.001	Grok > GPT > Gemini

**Table 6 diagnostics-15-01320-t006:** ROC analysis results and comparisons. GPT 0.5 = missing data replaced with intermediate values (0.5); GPT (+) = missing data considered positive; GPT (−) = missing data considered negative.

	AUC (95%CI)	vs. Gemini	vs. Grok	vs. GPT 0.5	vs. GPT (−)
Gemini	58.5% (57.9–59.1%)				
Grok	56.3% (55.7–57.0%)	z = 6.462, *p* < 0.001 *			
GPT 0.5	56.2% (55.5–56.9%)	z = 5.950, *p* < 0.001 *	z = 0.262, *p* = 0.793		
GPT (−)	56.1% (55.5–56.6%)	z = 8.427, *p* < 0.001 *	z = 0.867, *p* = 0.386	z = 0.733, *p* = 0.463	
GPT (+)	54.3% (53.7–55.0%)	z = 10.841, *p* < 0.001 *	z = 5.004, *p* < 0.001 *	z = 20.593, *p* < 0.001 *	z = 5.952, *p* < 0.001 *

* marks statistical significance

**Table 7 diagnostics-15-01320-t007:** Accuracy on pathological slices.

Pathology	Number of Positive Slices	GPT ^1^ Accuracy	Grok Accuracy	Gemini Accuracy	GPT Accuracy ^2^ (n)
HGG	1996	39.8%	57.7%	51.4%	48.9% (1625)
LGG	463	13.8%	30.0%	21.0%	18.8% (341)
Meningioma	2303	17.5%	36.1%	26.3%	24.6% (1644)
Schwannoma	408	10.8%	25.7%	15.0%	14.4% (305)
Cavernoma	481	12.9%	29.1%	16.2%	17.4% (356)
Metastatic Disease	520	20.0%	42.1%	31.5%	26.8% (388)
Other	323	8.7%	34.7%	27.9%	13.5% (207)
Post-op	503	11.9%	34.8%	24.1%	16.4% (366)
TOTAL	6997	22.3%	41.1%	32.1%	29.8% (5232)

^1^ Decline to answer is considered as wrong answer for GPT. ^2^ Only answered slices considered (n = number of answered slices).

**Table 8 diagnostics-15-01320-t008:** Accuracy on sequence-based analysis.

Subgroup	Number of Sequences	GPT Accuracy	Grok Accuracy	Gemini Accuracy
Normal	299	42.5%	0.3%	31.1%
HGG	236	92.8%	100.0%	97.9%
LGG	112	69.6%	92.9%	86.6%
Meningioma	548	70.6%	98.9%	85.6%
Schwannoma	140	67.9%	98.6%	86.4%
Cavernoma	193	70.5%	99.0%	77.7%
Metastatic Disease	72	79.2%	100.0%	91.7%
Other	66	74.2%	98.5%	87.9%
Post-op	127	66.1%	94.5%	85.8%
TOTAL	1793	68.7%	81.9%	77.7%

## Data Availability

A part of the investigated dataset was previously distributed as the Gazi Brains 2020 dataset [8]. The rest of the dataset is currently confidential. Additional data are available upon reasonable request.

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
