# Peer review of "Do LLMs Have ‘the Eye’ for MRI? Evaluating GPT-4o, Grok, and Gemini on Brain MRI Performance: First Evaluation of Grok in Medical Imaging and a Comparative Analysis"

_diagnostics, 2025, doi:10.3390/diagnostics15111320_

Round 1

Reviewer 1 Report

Comments and Suggestions for Authors

Reviewer Report for Manuscript ID: diagnostics-3650937

Title: Do LLMs Have ‘the Eye’ for MRI? Evaluating GPT-4o, Grok, and Gemini on Brain MRI Performance: A Comparative Analysis

General Assessment:

This manuscript presents a large-scale comparative evaluation of GPT-4o, Grok, and Gemini on brain MRI interpretation. The inclusion of xAI’s Grok, evaluated for the first time in this context, adds novelty and scientific value. However, important issues regarding reproducibility, task design, and clinical realism must be addressed. The binary classification task oversimplifies diagnostic decision-making, and the absence of metadata about model versions and inference settings hinders reproducibility. I recommend a major revision.

Major Comments:

  1. Missing LLM Version and API Metadata

    The manuscript does not specify the exact versions or access method (e.g., API or UI) for GPT-4o and Gemini, nor the inference parameters used (temperature, top_p, max_tokens). This is essential for reproducibility. The authors should clearly state all model versions, configurations, and evaluation dates.

  2. Oversimplified Pathology Prediction Task

    The study uses a binary classification (“abnormal” vs “normal”) rather than assigning a specific diagnosis. This formulation lacks clinical depth and risks inflating performance. A multi-class task (e.g., meningioma, glioma, metastasis, post-op, normal) would better reflect real-world diagnostic needs. At a minimum, the limitation of binary labeling should be clearly acknowledged.

  3. GPT-4o Refusal Rate

    GPT-4o declined to respond in approximately 28% of cases. While the authors applied post-hoc adjustments, the underlying cause and its implications for model usability and fairness should be explored in more depth.

  4. Lack of Visual Examples

    No sample images or predicted outputs are included. At least a few representative examples (true positive, false negative, model disagreements) should be presented to enhance clarity and interpretability.

  5. Low AUC Performance

    Reported AUC values (54–60%) are close to the chance level. The authors should discuss whether this reflects model limitations or flaws in task design (e.g., limited context, binary labels, single-slice input).

Minor Comments:

  • The use of single slices without adjacent context limits clinical realism and should be discussed.

  • Only T1, T2, FLAIR, and T1c axial images were used; modalities like DWI, STIR, MRA, and orientations like sagittal or coronal were excluded.

  • Distinguishing FLAIR and T2 is anatomically dependent; this should be noted.

  • Please clarify why T1 and T1c were treated interchangeably.

  • Consider revising the title to emphasize Grok’s novelty (e.g., “First Evaluation of Grok…”).

  • Describe image preprocessing steps (e.g., JPEG quality, resolution, base64 encoding).

  • Minor grammar and formatting issues remain, especially in the Discussion.

Recommendation: Major Revision

Comments on the Quality of English Language

The overall English is understandable, but there are minor issues in clarity, grammar, and structure—especially in the Discussion section. Improving sentence flow and avoiding ambiguous terms would help better convey the authors’ insights. A professional language editing service is recommended prior to publication.

Author Response

General Assessment:
This manuscript presents a large-scale comparative evaluation of GPT-4o, Grok, and Gemini on brain MRI interpretation. The inclusion of xAI’s Grok, evaluated for the first time in this context, adds novelty and scientific value. However, important issues regarding reproducibility, task design, and clinical realism must be addressed. The binary classification task oversimplifies diagnostic decision-making, and the absence of metadata about model versions and inference settings hinders reproducibility. I recommend a major revision.

Authors thank the reviewer for taking the time to carefully evaluate our study and make detailed comments.

Major Comments:

Comment 1. Missing LLM Version and API Metadata
The manuscript does not specify the exact versions or access method (e.g., API or UI) for GPT-4o and Gemini, nor the inference parameters used (temperature, top_p, max_tokens). This is essential for reproducibility. The authors should clearly state all model versions, configurations, and evaluation dates.

Response 1: Authors thank the reviewer for their comments. Exact dates of the questioning ("between 20 November 2024 and 10 December 2024") were already present in the methods section, lines 97–98 of the manuscript. Exact builds were stated by adding:

“The builds used were as follows: GPT-4o (gpt-4o-2024-08-06, released August 6, 2024), Gemini (gemini-1.5-pro-002, released September 24, 2024), and Grok (grok-vision-beta-0.1.0, released November 23, 2024).” (Lines 99–101)

The only parameter used was "max_tokens," which was already present as the last sentence of Appendix A. Appendix A clearly demonstrates all communication with the LLMs, and the rest of the code simply involved calling images and saving answers to text files. These do not affect reproducibility. Every variable relevant to reproducibility is already present in Appendix A.

Comment 2: Oversimplified Pathology Prediction Task
The study uses a binary classification (“abnormal” vs “normal”) rather than assigning a specific diagnosis. This formulation lacks clinical depth and risks inflating performance. A multi-class task (e.g., meningioma, glioma, metastasis, post-op, normal) would better reflect real-world diagnostic needs. At a minimum, the limitation of binary labeling should be clearly acknowledged.

Response 2: Authors thank the reviewer for their comments. This limitation was clearly acknowledged by adding:

“Another limitation was the binary classification system. Since not all pathologies are clearly visible on every MRI sequence, the only reliable question for single-slice analysis was whether the slice was normal or abnormal. We addressed this limitation by grouping slices according to the specific pathology present and assessing them separately.” (Lines 321–324)

Comment 3: GPT-4o Refusal Rate
GPT-4o declined to respond in approximately 28% of cases. While the authors applied post-hoc adjustments, the underlying cause and its implications for model usability and fairness should be explored in more depth.

Response 3: Authors thank the reviewer for their comments. The underlying cause was already discussed in the Discussion section, lines 314–317. We did not apply only one type of adjustment; various scenarios, both favoring and penalizing GPT, were presented and discussed separately.

Comment 4: Lack of Visual Examples
No sample images or predicted outputs are included. At least a few representative examples (true positive, false negative, model disagreements) should be presented to enhance clarity and interpretability.

Response 4: Authors thank the reviewer for their comments. Figure 2 already shows the agreement and disagreement rates among models. Selectively presenting a few images in the article would introduce severe selection bias and raise questions about the neutrality of the authors. However, 100 of the MRIs used with all sequences are publicly available, as mentioned in the article, and any reader may review those consecutively selected examples without any selection bias from our end.

Comment 5: Low AUC Performance
Reported AUC values (54–60%) are close to the chance level. The authors should discuss whether this reflects model limitations or flaws in task design (e.g., limited context, binary labels, single-slice input).

Response 5: Authors thank the reviewer for their comments. The following was added to clarify:

“No clinical information about patients was provided to any of the LLMs, as this study aimed solely to evaluate MRI interpretation capabilities rather than comprehensive clinical decision-making. This may have negatively impacted performance across all models. The importance of presenting clinical context may be assessed in dedicated studies.” (Lines 325–328)

Minor Comments:

  • The use of single slices without adjacent context limits clinical realism and should be discussed.
    Response: Addressed in Response 5.
  • Only T1, T2, FLAIR, and T1c axial images were used; modalities like DWI, STIR, MRA, and orientations like sagittal or coronal were excluded.
    Response: The most common sequences obtained worldwide were included in the dataset and the study. It is not feasible to investigate every sequence or orientation with such large numbers in a single study.
  • Distinguishing FLAIR and T2 is anatomically dependent; this should be noted.
    Response: The authors were unable to fully understand this comment. We would be happy to respond if the reviewer could kindly clarify further.
  • Please clarify why T1 and T1c were treated interchangeably.
    Response: “...since distinguishing them simply by assessing a single slice is not always possible.” Added to lines 113–114.
  • Consider revising the title to emphasize Grok’s novelty (e.g., “First Evaluation of Grok…”).
    Response: This novelty was mentioned in the Discussion section; however, the authors feel that emphasizing this in the title might impact neutrality.
  • Describe image preprocessing steps (e.g., JPEG quality, resolution, base64 encoding).
    Response: JPEG quality and resolution varied because all images were converted from their original voxel resolutions.
  • Minor grammar and formatting issues remain, especially in the Discussion.
    Response: Improvements were made throughout the text.

Reviewer 2 Report

Comments and Suggestions for Authors
  1. While the study benefits from a large number of MRI slices, the analysis is conducted solely at the slice level. For greater clinical relevance, the authors are encouraged to aggregate predictions at the patient level, for example using majority voting or weighted scoring per case.
  2. The refusal behavior of GPT-4o is a key outcome, yet the reasons for refusal are not systematically analyzed. A categorized summary of the refusal types (e.g., ethical disclaimers, format violations, ambiguous content) would improve the transparency and interpretability of the model’s behavior.
  3. The decision to treat T1 and T1c sequence labels as interchangeable in evaluation is questionable and could affect result reliability. The authors should provide a clear clinical rationale for this assumption or alternatively report separate performance metrics for these sequences.
  4. Only a single prompt format is used to query each LLM. The robustness of the models to small prompt variations is not assessed. Including an analysis of how prompt phrasing affects the outputs would provide insight into model stability and real-world usability.
  5. The study thoroughly compares models using standard metrics, but lacks an error pattern analysis. Identifying common failure cases—such as which pathologies or sequences are most frequently misclassified—could help guide future improvements or model fine-tuning strategies.
  6. Although the dataset is introduced in detail, there is limited discussion regarding its generalizability. Since the data comes from a single institution, the authors should acknowledge potential limitations in terms of population diversity or scanner variability.
  7. The paper concludes that LLMs underperform compared to dedicated models. However, it would be helpful to briefly comment on realistic future use cases (e.g., triage support, preliminary reads in low-resource settings) where such models might still offer value despite limited accuracy.
  8. The evaluation strategy considers non-responses from GPT as incorrect predictions in some analyses but excludes them in others. While this dual approach is informative, the interpretation of comparative results would benefit from a more consistent framing or a clearer justification for each scenario.
  9. The impact of model memory resetting between sessions is mentioned but not elaborated. Since real-world applications might benefit from contextual memory, the authors could discuss how this limitation might be addressed in future implementations.
  10. The study would be improved by a more detailed reproducibility statement. While parts of the dataset are publicly available, it is unclear whether code, prompt templates, or experimental settings can be shared for replication purposes.

Author Response

  1. While the study benefits from a large number of MRI slices, the analysis is conducted solely at the slice level. For greater clinical relevance, the authors are encouraged to aggregate predictions at the patient level, for example using majority voting or weighted scoring per case.

Response: Authors thank the reviewer for their comments. Since different pathologies are best visualized in different sequences, we aggregated slices belonging to the same sequences of the same patients, as requested by the reviewer. These additional results are now presented under the newly added subheading "3.8. Sequence-Level Analysis" and discussed between the lines 274–297.

  1. The refusal behavior of GPT-4o is a key outcome, yet the reasons for refusal are not systematically analyzed. A categorized summary of the refusal types (e.g., ethical disclaimers, format violations, ambiguous content) would improve the transparency and interpretability of the model’s behavior.

Response: Authors thank the reviewer for their comments. As explained in the manuscript, “Reasons to refuse varied from ethical claims to confessing inability to decide (differing liberally between attempts, even for repeat attempts of the same slice), therefore a systematic analysis of this refusal was not possible due to the probabilistic nature of this phenomenon” (Lines 146–150). Exact numbers and alternative approaches, both favoring and penalizing GPT, were presented clearly to improve transparency and interpretability. Additionally, a possible mechanism is discussed in lines 314–317.

  1. The decision to treat T1 and T1c sequence labels as interchangeable in evaluation is questionable and could affect result reliability. The authors should provide a clear clinical rationale for this assumption or alternatively report separate performance metrics for these sequences.

Response: Authors thank the reviewer for their comments. This was clarified by adding: “…since distinguishing them simply by assessing a single slice is not always possible” (Lines 113–114). Additionally, the numbers accepted as correct due to this interchangeability were explicitly provided for all LLMs individually at lines 159–160, 183–184, and 201–203.

  1. Only a single prompt format is used to query each LLM. The robustness of the models to small prompt variations is not assessed. Including an analysis of how prompt phrasing affects the outputs would provide insight into model stability and real-world usability.

Response: Authors thank the reviewer for their comments. Dozens of different prompts could be constructed, and each LLM may respond differently to each of them. It is not feasible to manually obtain and process data for 35,711 slices multiplied by three different LLMs and multiple prompt variations. Prompt engineering is a separate topic that could be explored in future studies specifically designed for that purpose.

  1. The study thoroughly compares models using standard metrics, but lacks an error pattern analysis. Identifying common failure cases—such as which pathologies or sequences are most frequently misclassified—could help guide future improvements or model fine-tuning strategies.

Response: Authors thank the reviewer for their comments. Since the primary aim of this study was to perform a large-scale performance comparison, it was not practically feasible to request reasoning for each answer and manually evaluate them. Dedicated, smaller-scale studies might be better suited to address such specific questions.

  1. Although the dataset is introduced in detail, there is limited discussion regarding its generalizability. Since the data comes from a single institution, the authors should acknowledge potential limitations in terms of population diversity or scanner variability.

Response: Authors thank the reviewer for their comments. The dataset contains images collected from multiple centers across the country, and the main center itself has three different MRI scanners (both 1.5 and 3 Tesla). This diversity is intended as one of the dataset’s strengths, but discussing that extensively in this paper would not be appropriate. If further questions regarding the dataset’s properties arise, 100 of the MRIs are currently publicly available in the Synapse repository for independent review, as indicated by the cited DOI.

  1. The paper concludes that LLMs underperform compared to dedicated models. However, it would be helpful to briefly comment on realistic future use cases (e.g., triage support, preliminary reads in low-resource settings) where such models might still offer value despite limited accuracy.

Response: Authors thank the reviewer for their comments. The following was added: “Subsequently, LLMs may prove very useful as easy-to-use triage or decision support systems once their reliability is established” (Lines 369–371).

  1. The evaluation strategy considers non-responses from GPT as incorrect predictions in some analyses but excludes them in others. While this dual approach is informative, the interpretation of comparative results would benefit from a more consistent framing or a clearer justification for each scenario.

Response: Authors thank the reviewer for their comments. Due to the nature of the statistical tests used, we had to select a binary option, and thus both possibilities were investigated. Common errors and correct answers are critical for Cochran’s Q and McNemar’s tests, so we could not simply assign correct and incorrect answers arbitrarily or in a presumed proportion. We applied an intermediate value (of 0.5) for ROC analysis alongside the all-positive and all-negative scenarios. If the reviewer has specific directions or scientifically valid suggestions, the authors would gladly add those analyses.

  1. The impact of model memory resetting between sessions is mentioned but not elaborated. Since real-world applications might benefit from contextual memory, the authors could discuss how this limitation might be addressed in future implementations.

Response: Authors thank the reviewer for their comments. Memory resetting is the default behavior for these models. Maintaining memory requires additional effort and processes. Thus, this is not a limitation of our study, but rather of the current default behavior of the models themselves.

  1. The study would be improved by a more detailed reproducibility statement. While parts of the dataset are publicly available, it is unclear whether code, prompt templates, or experimental settings can be shared for replication purposes.

Response: Authors thank the reviewer for their comments. All relevant code handling communication with the LLMs is presented in Appendix A, including the role assigned and the exact prompt used. As stated in the manuscript, it is not possible to share the entire dataset at present, as it is still a work in progress. However, a large sample (100 complete MRI scans with full segmentations) is publicly available at the cited DOI.

Regarding experimental settings, the following was added explicitly: “The builds used were as follows: GPT-4o (gpt-4o-2024-08-06, released August 6, 2024), Gemini (gemini-1.5-pro-002, released September 24, 2024), and Grok (grok-vision-beta-0.1.0, released November 23, 2024)” (Lines 99–101).

Reviewer 3 Report

Comments and Suggestions for Authors

This paper evaluates three LLMs for analyzing MRI scans of the brain. Despite the relevance of the topic, there are a number of issues that need to be addressed in this paper.

  1. The paper mainly details the results from existing LLM models and does not present any significant advances.
  2. The handling of missing data is problematic. It may not be appropriate to consider unanswered queries in ChatGPT-4o as incorrect.
  3. The use of the private Gazi Brains 2020 dataset raises questions about the representativeness of the data. It is difficult to assess whether the sample is sufficiently diverse to draw generalized conclusions about the performance of LLM in brain MRI interpretation.
  4. The authors should perform replicate experiments to eliminate changes in LLM performance over time.
  5. Selection criteria for pathology slices are not well defined and may lead to subjectivity.
  6. Considering T1 and T1c sequence predictions as “interchangeably correct” lacks clinical justification and artificially inflates the accuracy of sequence identification.
  7. Publicly disclose the code snippets used to invoke the LLM APIs.

Author Response

This paper evaluates three LLMs for analyzing MRI scans of the brain. Despite the relevance of the topic, there are a number of issues that need to be addressed in this paper.

  1. The paper mainly details the results from existing LLM models and does not present any significant advances.

Response: Authors thank the reviewer for their comments.

  1. The handling of missing data is problematic. It may not be appropriate to consider unanswered queries in ChatGPT-4o as incorrect.

Response: Authors thank the reviewer for their comments. Multiple solutions were applied to address this problem, including assigning intermediate values and considering all unanswered responses as either positive or negative. If the reviewer has specific directions or scientifically valid suggestions, the authors would gladly add those analyses.

  1. The use of the private Gazi Brains 2020 dataset raises questions about the representativeness of the data. It is difficult to assess whether the sample is sufficiently diverse to draw generalized conclusions about the performance of LLM in brain MRI interpretation.

Response: Authors thank the reviewer for their comments. The original Gazi Brains 2020 dataset is publicly available; however, an expanded version was used for the purposes of this study. As stated in the manuscript, it is not possible to share the entire dataset currently, as it remains a work in progress. However, a large sample (100 complete MRI scans with full segmentations) is publicly available at the cited DOI.

  1. The authors should perform replicate experiments to eliminate changes in LLM performance over time.

Response: Authors thank the reviewer for their comments. LLMs are probabilistic, not deterministic; thus, repeat attempts will not yield identical results. However, these minor deviations do not significantly impact the overall conclusions when investigating as many as 35,711 slices. Additionally, this inherent variability is precisely the rationale behind performing statistical analyses.

  1. Selection criteria for pathology slices are not well-defined and may lead to subjectivity.

Response: Authors thank the reviewer for their comments. This was already clearly stated in the manuscript lines 84–87: “Slices that contain contrast-enhancing or non-enhancing parts of a tumor or lesion, peritumoral edema, acute ischemia, encephalomalacic areas, cyst associated with a tumor, blood, or postoperative cavities were considered pathological and classified as ground-truth positive.” Additionally, reviewers and readers are welcome to inspect 100 public MRIs available at the Synapse repository if further questions arise regarding the dataset.

  1. Considering T1 and T1c sequence predictions as “interchangeably correct” lacks clinical justification and artificially inflates the accuracy of sequence identification.

Response: Authors thank the reviewer for their comments. This was clarified by adding: “…since distinguishing them simply by assessing a single slice is not always possible” (Lines 113–114). Additionally, the exact numbers of answers accepted as correct due to this interchangeability were explicitly reported for each LLM individually at lines 159–160, 183–184, and 201–203.

  1. Publicly disclose the code snippets used to invoke the LLM APIs.

Response: Authors thank the reviewer for their comments. Appendix A clearly demonstrates all communication with the LLMs. The rest of the code simply involved calling images and saving answers to text files, and thus is not relevant to reproducibility as it can be implemented in numerous ways without impacting the outcomes. Every variable relevant to reproducibility is already provided in Appendix A.

Round 2

Reviewer 1 Report

Comments and Suggestions for Authors Thank you for your thoughtful and comprehensive revisions. The revised manuscript demonstrates significant improvements in transparency and reproducibility. The following positive changes are noted and appreciated: •You have clearly specified the model versions, access dates, and querying protocol (including max_tokens) in the Methods section and Appendix A. •The discussion of GPT-4o’s refusal behavior is now more thorough, including its probabilistic nature and potential safety mechanisms. •The binary classification limitation is acknowledged in the Discussion, and the justification for its use in a single-slice context is reasonable. •References to prior studies on GPT-4 and Gemini in medical image analysis are appropriate and help contextualize your work.   That said, several minor but important issues remain and should be addressed or clarified to further strengthen the manuscript:   1. Multi-class classification should be acknowledged as a future direction. While you justify binary classification for this initial benchmark, the dataset clearly includes specific diagnostic labels (e.g., meningioma, glioma, metastasis, post-op). A brief comment in the Discussion or Conclusion that future studies could leverage multi-class classification for more clinically meaningful evaluation would be valuable.   2. Visual outputs would improve interpretability. While Figure 2 shows classification overlap among models, it does not reveal how or why the models succeeded or failed. Even a single representative slice with model predictions (TP, FN, or disagreement case) in a Supplementary Figure would improve transparency without introducing bias.   3. ROC curve interpretation should be moderated. Although you report statistically significant AUC differences between models, all AUC values are below 60%, and the ROC curves are near-linear. This indicates weak discriminative power. Please clarify in the Discussion that while comparative differences exist, overall performance remains low and should be interpreted cautiously.   4. Precision-Recall (PR) curves suggest class imbalance. The PR curves drop sharply at higher recall levels, suggesting that high sensitivity comes at the cost of poor precision. This likely reflects class imbalance (normal vs. abnormal = ~4:1). A brief mention of this limitation would be helpful.   5. Appendix A lacks full API configuration. While the base prompt and max_tokens are provided, please also include temperature, top_p, frequency_penalty, and presence_penalty values if available. These can affect response determinism and model behavior.   6. Image preprocessing is insufficiently described. Please clarify whether the images were resized, normalized, or compressed (e.g., JPEG quality level). These factors may influence model performance, particularly for subtle lesions.   7. Clarify the “FLAIR vs. T2” issue. The reviewer’s original concern was that distinguishing FLAIR from T2 may depend heavily on the anatomical level (e.g., presence or absence of visible CSF). Acknowledge that such cases may inherently limit model performance even in human readers.   8. Consider revising the title to highlight the novelty of Grok’s evaluation. While neutrality is important, including a phrase such as “First Evaluation of Grok” would better capture the unique contribution of your study and improve discoverability.   With these clarifications and minor additions, the manuscript will be suitable for publication and will serve as a valuable reference for future work on large language models in medical imaging. Comments on the Quality of English Language

The overall English is understandable, but some parts of the Discussion and Results would benefit from improved clarity and precision (e.g., consistent tense use, clearer transitions, avoiding vague terms like “performed well”).

Author Response

Thank you for your thoughtful and comprehensive revisions. The revised manuscript demonstrates significant improvements in transparency and reproducibility. The following positive changes are noted and appreciated:

  • You have clearly specified the model versions, access dates, and querying protocol (including max_tokens) in the Methods section and Appendix A.
    • The discussion of GPT-4o’s refusal behavior is now more thorough, including its probabilistic nature and potential safety mechanisms.
    • The binary classification limitation is acknowledged in the Discussion, and the justification for its use in a single-slice context is reasonable.
    • References to prior studies on GPT-4 and Gemini in medical image analysis are appropriate and help contextualize your work.

Response: Authors thank the reviewer for their comments. Also, as per the reviewer’s request language has been revised and various changes addressing subject verb disagreement issues and ambiguous statements have been made throughout the text.

That said, several minor but important issues remain and should be addressed or clarified to further strengthen the manuscript:

  1. Multi-class classification should be acknowledged as a future direction. While you justify binary classification for this initial benchmark, the dataset clearly includes specific diagnostic labels (e.g., meningioma, glioma, metastasis, post-op). A brief comment in the Discussion or Conclusion that future studies could leverage multi-class classification for more clinically meaningful evaluation would be valuable.

Response: Authors thank the reviewer for their comments. Acknowledging this, the following statement has been added to the Conclusion section (lines 391–393):
"Future studies could leverage multi-class classification to provide a more clinically meaningful evaluation beyond the binary classification approach used here."

  1. Visual outputs would improve interpretability. While Figure 2 shows classification overlap among models, it does not reveal how or why the models succeeded or failed. Even a single representative slice with model predictions (TP, FN, or disagreement case) in a Supplementary Figure would improve transparency without introducing bias.

Response: Authors thank the reviewer for their comments. Representative MRI slices illustrating model predictions have been included as Appendix B to improve transparency.

  1. ROC curve interpretation should be moderated. Although you report statistically significant AUC differences between models, all AUC values are below 60%, and the ROC curves are near-linear. This indicates weak discriminative power. Please clarify in the Discussion that while comparative differences exist, overall performance remains low and should be interpreted cautiously.

Response: Authors thank the reviewer for their comments. The following clarifying statement has been added to the Discussion section (lines 327–329):
"It should be noted that although differences between models were statistically significant, all AUC values remained relatively low (below 60%), indicating limited overall discriminative power."

  1. Precision-Recall (PR) curves suggest class imbalance. The PR curves drop sharply at higher recall levels, suggesting that high sensitivity comes at the cost of poor precision. This likely reflects class imbalance (normal vs. abnormal = ~4:1). A brief mention of this limitation would be helpful.

Response: Authors thank the reviewer for their comments. To acknowledge this limitation, we have added the following sentence to the Discussion section (lines 329–331):
"The sharp drop in Precision-Recall curves at higher recall levels likely reflects class imbalance, given that normal slices outnumber abnormal ones by approximately four to one."

  1. Appendix A lacks full API configuration. While the base prompt and max_tokens are provided, please also include temperature, top_p, frequency_penalty, and presence_penalty values if available. These can affect response determinism and model behavior.

Response: Authors thank the reviewer for their comments. The following clarification has been added to 2.2 Questioning LLMs subheading (lines 97–98):
" Other inference parameters were not explicitly set and were thus left to each model’s default behavior."

  1. Image preprocessing is insufficiently described. Please clarify whether the images were resized, normalized, or compressed (e.g., JPEG quality level). These factors may influence model performance, particularly for subtle lesions.

Response: Authors thank the reviewer for their comments. The following preprocessing detail has been added to the Methods section (lines 93-94):
"Image conversion to JPEG was performed using the OpenCV library with a quality setting of 95, in their original voxel resolution …"

  1. Clarify the “FLAIR vs. T2” issue. The reviewer’s original concern was that distinguishing FLAIR from T2 may depend heavily on the anatomical level (e.g., presence or absence of visible CSF). Acknowledge that such cases may inherently limit model performance even in human readers.

Response: Authors thank the reviewer for their comments. To acknowledge this nuance clearly, we added the following sentence to the Discussion section (lines 307–311):
"Although, distinguishing FLAIR from T2 sequences may depend heavily on the anatomical region being imaged, particularly concerning the visibility of cerebrospinal fluid, which poses a potential limitation for both human readers and automated models."

  1. Consider revising the title to highlight the novelty of Grok’s evaluation. While neutrality is important, including a phrase such as “First Evaluation of Grok” would better capture the unique contribution of your study and improve discoverability.

Response: Authors thank the reviewer for their comments. Title has been revised as:
"Do LLMs Have ‘the Eye’ for MRI? Evaluating GPT-4o, Grok, and Gemini on Brain MRI Performance: First Evaluation of Grok in Medical Imaging and a Comparative Analysis"

Reviewer 3 Report

Comments and Suggestions for Authors

The author has provided a good response to the questions raised and is recommended for publication.

Author Response

The author has provided a good response to the questions raised and is recommended for publication.

Response: Authors thank the reviewer for their comments and spared their valuable time to make recommendations to improve our manuscript.